# Future Prospects of Gene Therapy for Friedreich’s Ataxia

**DOI:** 10.3390/ijms22041815

**Published:** 2021-02-11

**Authors:** Gabriel Ocana-Santero, Javier Díaz-Nido, Saúl Herranz-Martín

**Affiliations:** 1Centro de Biología Molecular Severo Ochoa (CSIC-UAM), Departamento de Biología Molecular, Universidad Autónoma de Madrid, Nicolás Cabrera 1, 28049 Madrid, Spain; gabriel.ocanasantero@seh.ox.ac.uk (G.O.-S.); javier.diaznido@uam.es (J.D.-N.); 2Department of Physiology, Anatomy and Genetics, Sherrington Building, Parks Road, University of Oxford, Oxford OX1 3PT, UK

**Keywords:** gene therapy, neurodegeneration, Friedreich’s Ataxia, AAV, mouse models, preclinical studies, clinical trials

## Abstract

Friedreich’s ataxia is an autosomal recessive neurogenetic disease that is mainly associated with atrophy of the spinal cord and progressive neurodegeneration in the cerebellum. The disease is caused by a GAA-expansion in the first intron of the frataxin gene leading to a decreased level of frataxin protein, which results in mitochondrial dysfunction. Currently, there is no effective treatment to delay neurodegeneration in Friedreich’s ataxia. A plausible therapeutic approach is gene therapy. Indeed, Friedreich’s ataxia mouse models have been treated with viral vectors en-coding for either FXN or neurotrophins, such as brain-derived neurotrophic factor showing promising results. Thus, gene therapy is increasingly consolidating as one of the most promising therapies. However, several hurdles have to be overcome, including immunotoxicity and pheno-toxicity. We review the state of the art of gene therapy in Friedreich’s ataxia, addressing the main challenges and the most feasible solutions for them.

## 1. Introduction

Friedreich’s Ataxia (FA) is the most frequent of the autosomal recessive cerebellar ataxias. It is also the most common ataxia in Caucasians (two–five patients/100,000 individuals) [1]. FA is a systemic disease, in which the central nervous system (CNS) is severely affected. The main neurological symptoms include gait instability, loss of coordination in arms and legs, dysarthria, hearing loss, nystagmus, vision impairment, and mild olfactory dysfunction. Extra neurological symptoms may include muscle weakness, musculoskeletal abnormalities like pes cavus or scoliosis (55–90% of patients), hypertrophic cardiomyopathy (60–70% of patients), and diabetes type I (20–30% of patients) [1]. Within the CNS, cerebellum, and spinal cord are the most affected areas. The atrophy of the dorsal horn in the spinal cord is the earliest event [2], taking place in early childhood, so it is thought to be a neurodevelopmental disorder. In later stages, a progressive neurodegeneration is extended to the cerebellum, especially in the dentate nucleus [3]. The onset of the disease is during the first decades of life. The progression of the disease may confine the patient to a wheelchair 15–20 years after the onset [1].

The disease is caused by recessive mutations in the gene encoding for the protein Frataxin (FXN). This protein is mainly localized in the mitochondria [4] and it plays an important role in iron metabolism, since it is involved in the biogenesis, maintenance, and repair of Fe-S clusters [5]. The most common mutation is an expansion of the triplet GAA in the first intron of the *FXN* gene. The expansion causes the epigenetic silencing of the gene. A total of more than 30 repeats is considered pathogenic and patients usually have 600–900 repeats [6]. Moreover, the expression of FXN and the onset of the disease is related to the number of repeats, meaning that a high number of repetitions implies a reduction in the level of FXN (5–35% in comparison to normal levels) and causes an earlier onset of the disease. Although it is still not fully understood how this process leads to neurodegeneration, the current hypothesis suggests that the impairment of the Fe–S clusters leads to mitochondrial dysfunction that eventually triggers apoptosis [7].

One of the main challenges for the study of FA has been to develop animal models to recreate the disease. The first attempt was to generate a knockout of *Fxn*, however it is lethal during embryonic development [8]. The current tendency is to use *Fxn*^−/−^ mice bearing a human *FXN* transgene with GAA triplet expansions, such as the YG8R [9] or the YG8sR [10] models with repeats ranging from 90 to 200. Over the last few months a new mouse model has been under investigation, Fxn^null^::YG8s(GAA)_>800_ (herein referred to as YG8JR) that also lacks the murine *Fxn* and carries human *FXN* with around 800 repeats, which makes it genetically closer to human patients and, therefore, ideal to screen possible therapies.

Indeed, there is no cure yet, and treatments are only palliative [11]. For this purpose, different therapeutic approaches have been studied [11]. One of them is to face the mitochondrial dysfunction using, for instance, enhancers of energy metabolism (L-carnitine and creatine) [12] or antioxidants (co-enzyme Q_10_ and its analogue idebenone), that will also stimulate mitochondrial activity [13]. Another strategy is to increase the levels of endogenous FXN with molecules such as erythropoietin (EPO) [14], immune modulators, (IFNγ-1b) [15] or epigenetic drugs [16]. However, one of the most promising approaches is gene therapy. In this context, different viral vectors are being studied, such as herpes simplex virus (HSV) [17,18] or adeno-associated virus (AAV), among others. Lately, AAV vectors have been widely used in studies for many neurodegenerative diseases [19]. In FA, previous studies already attempted to deliver FXN through AAV in an FA mouse model, showing less neurodegeneration and longer life span [20]. Among AAVs, serotypes 9 and rh10 have proved to be the best choice for CNS delivery, because of their high efficiency transducing neurons and astrocytes [21,22].

In this work, we review the use of viral vectors in gene therapy, routes of administration for CNS delivery, mouse models of FA, and the state of the art of gene therapy for FA.

## 2. In Vivo Gene Therapy and AAV Vectors

In the last decades, gene therapy has been consolidated as a promising treatment to many genetic disorders, such as spinal muscular atrophy (SMA) or Leber’s congenital amaurosis, with some gene therapy drugs already approved [23,24]. Other diseases with numerous preclinical and clinical studies are also being conducted, including, among others, Alzheimer’s disease, Amyotrophic lateral sclerosis (ALS), or Huntington’s disease [25,26,27,28]. This approach is based on the delivery of nucleic acids to patients in order to reverse or prevent their pathological phenotypes. In this context, there are several types of vectors that can be broadly classified in two major groups: non-viral and viral vectors. The major advantage of non-viral vectors is their extremely low immunotoxicity while the main disadvantage is their low delivery efficiency. This issue can be addressed by different delivery methods including physical approaches, such as electroporation, magnetofection, or ballistic DNA, among others; and chemical methods, including, for instance, gold nanoparticles, lipoplexes, or polyplexes [29]. Their high biosafety is leading to an increasing number of clinical trials with these vectors, for example liposome-mediated gene therapy for cancer treatment [30]**.** The improvement of these vectors and their delivery is under intensive study [31], being the use of extracellular vesicles, such as exosomes, one of the most promising gene transfer systems for neurological diseases [32]. Nevertheless, nowadays viral vectors are still a more realistic approach. Attenuated viruses are proving to be excellent systems for the delivery of nucleic acids due to their higher transduction efficiency.

Viral vectors profit the infectious ability of viruses to transduce cells while carrying the genes of interest. For this purpose, non-essential genetic material for transduction is removed from viruses, rendering them unable to replicate and nonpathogenic. In order to choose the appropriate viral vector, different features have to be taken under consideration (Table 1) [33]. First of all, some vectors, as retroviruses, only transduce dividing cells leaving them inadequate for diseases that affect non-dividing cells, like neurons. In contrast, non-integrative vectors such as HSV are rather poor vectors for the treatment of diseases affecting highly dividing-cells such as sickle cell disease (myeloid stem cells), since HSV-derived vectors are extra-chromosomal and may not reach all descending cells upon replication. In this context, AAVs can remain episomal, however some studies have also shown vector integration through both non-homologous sites and homologous recombination [34]. Overall, these vectors can achieve long-term expression and are being combined with CRISPR (AAV-based CRISPR systems) to study potential treatments for β-hemoglobinopathies like sickle cell disease (ex vivo gene therapy) [35]. Another additional concern is the size of the gene that we aim to deliver. While some vectors, such as HSV or Vaccinia can pack more than 30 kbp, others, such as AAV can only carry inserts of up to 4.5 kb. Other issues to consider are target tissue (e.g., blood–brain barrier (BBB) has to be crossed to reach CNS), immune response triggered, and host range and stability of expression among others. The major drawbacks of viral vectors are their potential to trigger immune responses in the patient and the risk of cancer associated with integrative vectors.

Gene therapy includes in vivo approaches, consisting of the direct delivery of the viral vector into the patient or ex-vivo, which involves transducing cells extracted from the patient in order to return them once modified. For the treatment of CNS, target cells are not only hard to access but also post-mitotic, making ex-vivo gene therapy approaches non-viable. Therefore, in vivo gene therapy seems to be the best strategy. In this context, AAV is the most promising vector and the vast majority of gene therapy clinical trials for neurodegenerative diseases use AAVs [28]. Furthermore, gene therapies using these viral vectors have already been approved by the U.S. Food and Drug Administration (FDA). For example, Luxturna (AAV2.*hRPE65v2)* for the treatment of Leber’s congenital amaurosis type 2 (LCA2), a retinal dystrophy that affects children [23] and Zolgensma, which prevents the early death (usually before the 2nd birthday) of patients with spinal muscular atrophy (SMA) by delivering *SMN1* through an AAV9 [44]. SMA patients are treated with Zolgensma no later than two years of age and the first patients have already reached five years of age [38].

AAV vectors have several advantages: they are really efficient transducing post-mitotic cells, trigger a moderate immune response at first exposure, and can go through the BBB. There is a variety of serotypes of AAVs with different antigens in their capsids (Table 2). Each serotype has a different tropism, and it is important to choose the appropriate serotype based on the target tissue. For CNS, AAV9 and AAVrh10 have shown the broadest biodistribution through the brain and spinal cord. Both are currently used in clinical trials in different neurological diseases, for example in a phase I trial in Alzheimer’s disease patients (AAVrh.10Hpoe2) [45] and a phase I trial in Giant Axonal Neuropathy (scAAV9/JeT-GAN) [46]. As previously mentioned, Zolgensma, a gene therapy with AAV9 targeting CNS, has already been approved by the FDA. Whether the best choice for CNS is AAV9 or AAVrh10 remains controversial, with studies defending a higher transgene expression with AAVrh10 [22], while others attribute a greater biodistribution to AAV9, because of axonal transport of its genome [47]. For the treatment of FA, both serotypes are extremely promising given their tropism towards the cerebellum, spinal cord, and heart.

As previously stated, the two major drawbacks of viral vectors are the mutational risk by insertion of the vector genome in the genomic DNA (genotoxicity) and the immune response that they may cause (immunotoxicity). Genotoxicity can be easily avoided by using non-integrative vectors such as AAV or HSV. AAVs also have a low insertional frequency, nevertheless, some studies indicate a potential tumorigenesis effect [66]. These results highlight the importance of studying the genotoxicity of specific AAVs, however, this does not directly affect the viability of these vectors since several publications have described safe non-tumorigenic AAVs [67]. Moreover, addressing immunotoxicity is challenging and extremely important because it can render the therapy non-viable, and/or endanger the life of the patient. The first approach to tackle this is to fine-tune the vector dose in order to find a dose that maximizes transgene expression while minimizing the immune response triggered. Then, to carry out immunomodulation strategies to avoid or reduce the immune response, such as the use of peri-procedural corticosteroids and immunosuppressive agents like rituximab and rapamycin [68]. Finally, the route of administration has to be considered to avoid transduction in off-target tissues, which will allow to minimize the dose required and, subsequently, diminish the immune response generated. Therefore, to perform and deeply analyze immunotoxicity studies is essential to succeed in gene therapy studies. For instance, toxicity has been recently reported in the dorsal root ganglion (DRG) sensory neurons [69] after AAV injection in non-human primates. According to this study, DRG pathology was more severe when higher dose vector or older animals were injected. In diseases already showing DRG neurodegeneration such as FA [2], this has to be taken into account.

## 3. Routes of Administration for Central Nervous System Delivery

In vivo gene therapy requires the direct administration of viral vectors into the patient. In this sense, selecting the appropriate route of administration is vital for a successful therapy. Depending on the delivery route, some differences are observed among transduction efficiency, immune-response, transgene biodistribution, and translational potential of a proposed therapy. Given the neurodegenerative nature of FA, we have focused on the delivery routes that are most suitable for transducing cells in the CNS, especially those involving AAV vectors that target the cerebellum.

The different administration routes of AAV for gene therapy (Table 3, Figure 1) have been deeply discussed elsewhere [68]. Systemic delivery is the most common approach for gene therapy of multi-systemic diseases (e.g., multiple sclerosis). This route has obvious advantages, clinically it is a common practice and it is the easiest procedure. Additionally, some diseases with a predominantly neurological phenotype are in fact systemic diseases (e.g., FA). In this sense, systemic delivery of a vector with a relatively ubiquitous promoter could solve the gene deficiency in all affected tissues. For instance, Zolgensma, the AAV9 gene therapy treatment for SMA is delivered through systemic administration [44]. The therapy requires a single 60-min intravenous infusion with 1.1 × 10^14^ v.g./Kg of body weight of AAV9 carrying the *SMN1* gene.

When targeting cells in the CNS, vectors must cross the BBB so high doses of the vector are required, which may increase the immune response generated. Still, most viral vectors are unable to cross the BBB, independently of the dose. AAVs can cross the BBB, but not all serotypes have the same efficiency, having most of the AAVs a rather poor efficiency. Nevertheless, AAV9 and AAVrh10 have shown a great capacity to go through the BBB [70]. Despite their higher efficacy, several strategies are under study to increase the crossing of the endothelial cell barrier. One of the most promising is the use of shuttle peptides (e.g., THR peptide) enhancing the transport of virions across the BBB [71]. Moreover, direct injection into the CNS allows bypassing the BBB and therefore has to be considered.

Direct delivery to CNS allows the injection of lower doses, so the immune response generated should be lower too. In this context, three main routes of injection have been used for AAVs: intrathecal (IT), intra-cisterna magna (ICM), and intracerebroventricular (ICV) (Figure 1a). CSF injection achieves a greater biodistribution through CNS, which is necessary for the treatment of diseases like FA where more than one region is severely affected [1,72]. Additionally, some procedures such as direct injection in the 4th ventricle [73] are only used at the preclinical level because of their higher invasiveness. Moreover, intraparenchymal injections are also used. Injection into the deep cerebellar nuclei [74] or into the cerebellar cortex [18] is employed in mice. Indeed, there are several AAV gene therapy clinical trials with intraparenchymal CNS delivery, for example, for the treatment of late infantile neuronal ceroid lipofuscinosis [75], Parkinson’s Disease [76], or Canavan Disease [77]. Intraparenchymal approaches assure great spatially-controlled transduction in a specific region of the CNS which is desirable in diseases like Parkinson’s Disease.

Lumbar IT injection is a common clinical practice for the administration of analgesic agents or chemotherapy. Regarding its use as a delivery route for gene therapy approaches, studies in cynomolgus macaques illustrate that despite the high transduction efficiency in the spinal cord and some organs, such as the heart and lungs, few viral genomes reach the cerebellum and other brain regions [58]. These results contrast with a recent study in mice describing an efficient transduction of deep brain regions [78]. Coherently with its high efficiency transducing spinal cord, IT injection is being used in preclinical assays for ALS, a neurodegenerative disease that mainly affects motor neurons. This approach is showing promising results in both mouse and cynomolgus macaque models [26]. Additionally, a phase I clinical trial using IT injection of AAV9 carrying GAN for the treatment of giant axonal neuropathy is underway since 2015 [79].

ICM injection consists of reaching the cerebrospinal fluid (CSF) through the cisterna magna, which is anatomically found under the 4th ventricle. ICM approaches have shown high expression of the transgene in the spinal cord, cerebellum, and even the brainstem of cynomolgus macaques [58]. A study using ICM injection in mice also described a high transduction efficiency in the brain, with some vector genomes reaching the peripheral nervous system. This study shows spinal cord transduction, although in lower efficiency than the IT injection [78]. Cortical structures also have detectable levels of the transgene of interest, however worse than through other approaches, such as ICV. The major disadvantage of this route is that it is not usually applied in clinical practice [68]. The main reason is the lack of a safe procedure, with common complications related to medullary injury [80]. Despite this fact, it is commonly used in preclinical studies and several groups are trying to analyze its toxicity [81] and to establish a standard protocol with the purpose of eventually applying it to clinical studies [80].

Finally, the bilateral ICV route is based on the direct delivery into the CSF through the lateral ventricles. ICV provides the greater biodistribution among CNS structures being also a safe and commonly used approach in clinical practice [82]. Nonetheless, it does imply some risks such as infections, increase of intracranial pressure, or intracranial hemorrhages. Luckily, these risks are associated with chronic procedures rather than single delivery. Thus, single AAV delivery should be safe. Indeed, several pre-clinical studies are using this kind of approach, for example, AAV2/9 carrying *NDUFS4* to treat Leigh syndrome [83]. Additionally, preclinical studies for the treatment of diseases with a strong cerebellum phenotype, such as Niemann Pick disease, are showing promising results by ICV delivery of AAV9.hSYN1.*hNPC1.*SV40pA [84].

## 4. Friedreich’s Ataxia Animal Models

Even though FA is considered a rare disease, it is the most common form of hereditary Ataxia in Caucasians. The relatively high incidence combined with the severe symptomatology and its notorious economic impact makes the search for a therapy essential. Despite the efforts made, there is no cure or effective treatment yet [11]. The lack of viable therapies is partially due to the absence of animal models that faithfully recreate the disease.

Several disease models have been developed over the last few years. A summary of the characteristics of the main mouse models is outlined in Table 4. To compel all the symptomatology of this multi-systemic disease in a single animal model is challenging. Although several different models of the disease have been produced in *Drosophila melanogaster* and *Caenorhabditis elegans,* the vast majority of preclinical studies in FA have been conducted in mouse models of the disease.

Consistent with other animal models, complete *Fxn* gene knockout (K.O.) in mice is embryonically lethal [8]. The first approaches to generate a viable mouse model used the Cre-recombinase system to decrease frataxin only in the heart or neurons [85] by controlling the recombinase expression under the muscle creatine kinase (MCK, muscle-specific) and neuron-specific enolase (NSE, neuron-specific) promoters, respectively. Both models presented cardiomyopathy and iron accumulation with an additional ataxic phenotype in the NSE model. Both models presented a notorious reduction of their life span. The ensuing strategies attempted to introduce GAA repeats in the endogenous *Fxn* sequence to recreate the disease more faithfully from a molecular and genetic perspective. This was done by homologous recombination using a 230-repeat GAA sequence cloned from a patient and inserted in the endogenous *Fxn* locus of the mouse. The first model was called KIKI (knock-in knock-in) and had homozygous *Fxn* (GAA)230 [87]. This model achieved an approximate decrease in frataxin levels of 25%. In order to accentuate this reduction, these mice were crossed with an *Fxn*^+/−^ mice to generate the KIKO (knock-in knock-out) model (*Fxn*(GAA)230^/−^) [87]. KIKO achieved a decrease of 70% in frataxin levels. Recent studies have found significant cellular and molecular changes, including deficits in mitochondrial biogenesis and respiratory chain complexes [88], as well as some synaptic alterations in the cerebellum of KIKO mice [89], which may be considered a model of an early phase of cerebellar pathogenesis in FA (before clinical manifestations). Additionally, a deeper examination of the behavioral performance of KIKO mice has described neurobehavioral deficits starting at nine months of age [90]. Thus, this model ratifies the usefulness of this model for the study of FA therapy efficacy.

At this point, a new strategy to develop FA mouse models emerged by using yeast artificial chromosomes (YAC) encoding human *FXN* with GAA repeats in *Fxn* K.O. mice [9]. Following this procedure, YG22R and YG8R were developed. YG22R has a YAC with an *FXN* copy carrying 190 GAA repeats, while YG8R carries two sets of repeats, one with 90 and the other with 190 repeats. These models achieved 50–60% expression of human frataxin in the cerebellum compared to the endogenous levels of FXN in WT mice. Both models present mild coordination and motor defects as well as a slight oxidative stress. Furthermore, some mitochondrial alterations similar to those found in KIKO mice have also been described in cerebellar neurons from YG8R mice [91]. However, their major success was to recreate for the first time the GAA-repeat instability that human patients show [92]. GAA instability leads to expansions and contractions in the number of GAA repeats from one generation to the next one. Following a similar approach, the YG8sR was produced. Nevertheless, this model bears only one set of 200 GAA repeats. YG8sR exhibits coordination, behavior, and motor deficits, as well as decreased aconitase activity and GAA instability. Even though all these models have a pathological phenotype, it is rather mild. One possible explanation could be the reduced number of repeats when compared to the genome of the human patients. This problem is addressed by the Fxn^null^::YG8s(GAA)_>800_ (YG8JR, https://www.jax.org/strain/030395), a mouse model lacking endogenous *Fxn* and expressing a human form of *FXN* with 800 GAA repeats through a YAC vector. This number of repeats makes the YG8JR a more faithful model of the disease from a genetic perspective, making it a promising candidate for preclinical research. Published data describing its phenotype are still missing, however some recent unpublished data point towards an ataxic phenotype.

## 5. Gene Therapy in Friedreich’s Ataxia

As previously mentioned, multiple approaches have been studied (e.g., antioxidants, epigenetic drugs, etc) for the treatment of FA, and gene therapy is one of the most promising. As opposed to the previously mentioned strategies, gene therapy aims to become a cure for the disease rather than a chronic treatment. Despite the lack of animal models mimicking FA, some successful attempts to implement gene therapy at a preclinical level have already been conducted (Table 5). The first one used LoxP[*FXN*] mice injected with HSV-Cre in the brainstem to knock out FXN in a spatially controlled region. Four weeks after the first injection, mice were injected with HSV1-*FXN.* The delivery of frataxin reverted the motor coordination impairment assessed by the Rotarod test [17]. Gene therapy approaches using AAV vectors have been used as well. Thus, the group headed by Helene Puccio treated MCK mice with AAV vectors encoding for *FXN* [93]. This study showed that intravenous injection of the vector prevented cardiomyopathy and, additionally, mice treated with the vector after the onset of the pathology fully reversed their heart failure. This effect correlated with a molecular prevention/recovery of mitochondrial organization, as well as Fe-S proteins that were deficient in the untreated mice. Subsequently, the same laboratory used a parvalbumin conditional K.O. of FXN to show that FXN delivery through an AAV vector leads to a recovery of sensory neuropathy in this model [86]. Furthermore, in studies conducted by other groups, MCK and NSE mice were treated with AAV9-*hFXN* by intraperitoneal injection [20]. The results showed that 6 × 10^11^ v.g. of the vector were enough to achieve a drastic improvement in life span, as well as cardiac function recovery. Finally, a recent study has described the recovery of cardiac performance in the αMyhc mouse, a stress-induced model of the cardiac pathological traits of FA, after intravenous injection of the vector AAVrh.10hFXN [94].

Additionally, in the context of gene therapy for the treatment of neurodegenerative diseases, neurotrophins are really promising candidates. Neurotrophins are small secretable proteins playing a pro-survival and anti-apoptotic role in the nervous system. As they are essential for neuronal survival, it has been proposed that their delivery through gene therapy could have the potential of arresting or slowing down neurodegeneration. Delivery of neurotrophins has already been attempted for the treatment of different neurodegenerative diseases, with encouraging results [95,96]. Among the different neurotrophins studied, one of the most promising for gene therapy approaches is BDNF. BDNF exerts a tripartite protective role, it is an anti-apoptotic, anti-oxidant, and anti-autophagy agent [97]. Furthermore, BDNF is secreted, which implies that not only the transduced cells will increase their BDNF levels, but also the neighboring cells. Moreover, it was recently demonstrated that an miRNA targeting BDNF is increased in fibroblasts from FA patients and that, coherently, BDNF levels are reduced [98]. Therefore, BDNF is a potential candidate for the treatment of neurodegenerative diseases, and especially for FA. BDNF delivery through AAV vectors has already been studied in neurodegenerative diseases like Alzheimer’s disease, showing a recovery in the neurodegenerative and behavioral phenotype of a mouse model of the disease [99]. BDNF is also being studied in ataxic mouse models such as *stargazer* mice. When this model was crossed with mice overexpressing transgenic BDNF, a prevention of ataxic and motor impairment was described [100]. Furthermore, it was also demonstrated that delivery of BDNF through a herpesviral amplicon vector in mice deficient in FXN led to a prevention in neurodegeneration. Frataxin-deficient mice were generated by injection in the cerebellar cortex of a lentiviral vector carrying an shRNA against *Fxn*. BDNF delivery rescues the neurodegenerative phenotype in the cerebellum, as well as the coordination impairment present in these mice [18].

A point of major concern in gene therapy is phenotoxicity or the toxicity associated with the overexpression of the transgene. Studies in cultured human cells have demonstrated that FXN overexpression also leads to oxidative stress and toxicity, in a similar way to frataxin deficiency, underscoring the importance of physiological levels of FXN to confer a therapeutic effect [101]. More recently, a study reports that a very high level of FXN overexpression leads to mitochondrial dysfunction and cardiac toxicity in mice [102]. According to this, FXN cardiac overexpression up to nine-fold of the normal endogenous level seems safe, but significant heart toxicity is observed above 20-fold [102]. Therefore, it is also important to check the level of FXN overexpression that is safe for neuronal function in the cerebellum and to control the threshold to avoid neurotoxicity. Similar considerations have to be made when considering the use of neurotrophins. Several studies report the toxic effects of an excessive amount of neurotrophins. For example, it has been reported that a mouse line overexpressing BDNF in the forebrain exhibits memory and learning deficits [103]. Altogether, these studies highlight the necessity to tune the levels of transgene overexpression achieved. We have already discussed how an appropriate vector serotype and delivery route will allow the administration of lower vector doses. Additionally, the design of the vector is crucial for a finely tuned transgene expression and biodistribution.

The first consideration is the promoter, which can be ubiquitous or cell-type specific. On one side, ubiquitous promoters will allow a broad expression through different cell types. The most commonly used are enhancer/chicken β-actin (CAG), cytomegalovirus (CMV), and the human elongation factor 1α (EF1α) promoters [104]. An example of CNS gene therapy using this type of promoter is Zolgensma, which carries a hybrid cytomegalovirus enhancer/chicken β-actin promoter [24,105]. Additionally, some preclinical gene therapy studies for FA have used the CAG promoter [86,93]. On the other side, cell-type-specific promoters allow to confine transgene expression to certain cell types, thus reducing off-target phenotoxicity. Their major drawback is their lower rate of expression and their frequently higher size. The most commonly used promoters for CNS gene therapy are synapsin-1 (Syn1) and neuron-specific enolase (NSE) promoters to target neurons and the glial fibrillary acidic protein (GFAP) promoter for astrocytes [104], which have been used in several pre-clinical studies [106]. For the treatment of FA, the use of neuronal promoters at a preclinical level would allow for a more precise study of neuropathy reversion in animal models of the disease. Moreover, the addition of regulatory sequences, such as introns, enhancers, silencers, polyadenylation sequences, and insulator elements can allow for a finer tune of the dose [104]. The use of the own transgene promoter or the regulation of its 3′UTR elements have already been examined when, for instance, the transgene is toxic. For example, this is the case in a preclinical assay for the treatment of Rett syndrome by delivering MECP2 through an AAV vector [107]. This is especially promising in FA gene therapy given the toxicity of FXN overexpression [102]. Indeed, the use of their own *FXN* regulatory sequences has already been tested using infectious bacterial artificial chromosomes (iBACs) carrying the entire *FXN* genomic locus in an HSV-1 system [108,109]. By using this approach, fibroblasts from FA patients were rescued from oxidative stress in vitro [108]. Moreover, in vivo results using this system showed stable expression of a reporter transgene under the control of the *FXN* native promoter up to 75 days after the injection of the vector into the adult mouse cerebellum [109].

Coherently, recent publications have attempted to shed some light on the regulation of FXN expression. A recent study of CIS transcriptional regulation has defined the minimal regions required for an efficient FXN expression, including sequences from the 5′UTR and the first intron [110]. In the same study, these sequences have been used to produce different mini*FXN* genes. These minigenes have an efficient expression in HEK293 cells, but also in induced pluripotent stem cells, induced neurons, and induced cardiomyocytes, after AAV-mediated transduction. Additionally, they showed an efficient expression in mice, after ICV injection of AAVs bearing the minigenes. Thus, these results provide strong evidence supporting the viability of using endogenous *FXN* sequences in FA gene therapy. Another recent study focused on putative epigenetic regulatory elements [111]. Conducting a bioinformatic analysis of data extracted from ENCODE and Roadmap consortia databases, the research group has proposed new potential epigenetic regulatory regions. In this study, the enhancers at the end of intron 4 are specially highlighted due to their weakness or absence in FA-affected tissues. A better understanding of FXN expression regulation could strongly improve vector design, leading to an appropriate transgene expression from a temporal, spatial, and quantitative perspective, thus bypassing phenotoxicity.

Finally, when considering FA gene therapy, the existence of several FXN isoforms should be taken into account. FXN I, the most extensively studied, is the canonical isoform and shows the mitochondrial location. However, two additional extra-mitochondrial isoforms have been described, being FXN II more abundant in CNS, including the cerebellum, and FXN III more related to the heart [112,113]. These isoforms have also shown different protective mechanisms when overexpressed in cell lines [112,113]. Altogether, the differences among these isoforms, lead to value the necessity of rescuing the expression of all of them in order to treat FA patients. In this regard, a study from 2015 demonstrated that the delivery of a 135 kb human FXN locus, through an HSV-1 vector, led to the overexpression of the three isoforms [114]. Further clarification on the physiological relevance of these isoforms is required but it seems clear that their differential expression must be considered when devising gene therapy treatments.

## 6. Discussion

Undoubtedly, fighting neurodegenerative diseases such as FA is going to be one of the biggest challenges of the 21st century. In this sense, one of the most promising approaches under study is gene therapy. This therapeutic strategy aims not only to treat diseases but rather to cure them. For gene delivery, many vectors have been analyzed. However, currently, the most realistic approach is to use attenuated viral vectors. In this sense, as previously discussed, AAVs excel because they trigger a low immune response at first exposure, can cross the BBB, remain episomal avoiding the risk of cancer by insertion, and have a great and stable transduction of post-mitotic cells such as neurons. There is a broad range of AAVs serotypes, with serotypes 9 and rh10 showing a great tropism towards neurons [21,22]. The potential of these vectors has been established by numerous preclinical and clinical studies, as well as by already approved therapies [38,44,45,46]. Routes of delivery have to be taken into account and based on the literature, ICV injection provides the broader expression through CNS structures [68]. Given the spinal cord pathology also observable in FA, we should consider that the route of administration must allow transgene expression in the spinal cord. Some publications describe overexpression in the spinal cord after AAV ICV injection [115]. However, for clinical implementation, intravenous systemic delivery is in fact the most used approach because of its extremely low invasiveness. Despite its lower efficiency delivering into the CNS, the validity of systemic delivery for CNS treatment has been established by Zolgensma [38]. Based on previous literature, FXN and BDNF delivery are promising approaches for the treatment of FA [17,18,20,86,93,94]. The fact that both proteins are reduced in cells from human patients further corroborates their potential. Nevertheless, several studies state that overexpression of FXN or BDNF may have toxic effects [102,103]. This highlights the necessity to tune vector dose, expression, and biodistribution. In this sense, understanding how FXN expression is regulated is crucial, as it will allow for designing vectors with the endogenous sequences, consequently decreasing the risk of phenotoxicity. Additionally, the lack of animal models that faithfully recreate the human disease also makes it challenging to analyze therapies on them. This leads to value the potential of the new YG8JR mouse model, which carries a human *FXN* containing 800 GAA repeats, making it the animal model most genetically similar to patients that has been developed so far. The phenotype of the YG8JR model is currently being investigated by various laboratories (data not published).

## 7. Conclusions

In this work, we reviewed the current state of viral vectors in gene therapy, routes of administration for CNS delivery, and mouse models of FA. Additionally, we have revised gene therapy approaches delivering FXN and/or BDNF to treat FA models. The viral vector immunotoxicity and the phenotoxicity associated with both frataxin and neurotrophins denote the necessity to optimize vector serotype, design, dose, and route of delivery. Finally, we highlight the therapeutic potential of FXN and BDNF delivery through AAV vector serotype 9 or rh10 for diseases such as FA and the potential of the YG8JR model to evaluate the viability of new therapeutic options.

## Figures and Tables

**Figure 1 ijms-22-01815-f001:**
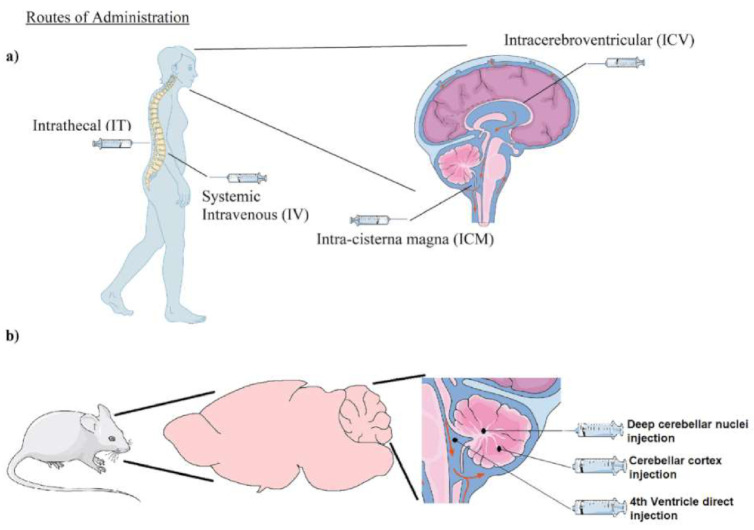
Different routes of administration used in gene therapy for CNS delivery. (**a**) Routes of delivery implemented clinically (IV, IT, ICV) or in the progress of being implemented (ICM). (**b**) Preclinical routes of gene delivery used in mice. IV: intravenous. IT: Intrathecal. ICV: Intracerebroventricular. ICM: Intra-cisterna magna.

**Table 1 ijms-22-01815-t001:** Viral vectors in gene therapy.

Vector	Genome	Insert Capacity	Expression	Immune Response	Features	Most Advanced Therapy Stage
Adenoviruses (Ad)	dsDNA	<7.5 kbp	Weeks/ Months Episomal	High	Infects both non- and dividing cells	Approved COVID19 [36,37]
Adeno-associated virus (AAV)	ssDNA	<4.5 kb	Years Episomal	Low	Long-term only in non-dividing cells	Approved Zolgensma [24,38]
Herpes simplex virus (HSV)	dsDNA	>30 kbp Up to 150 kbp	Months Episomal	High	Long-term only in non-dividing cells	Phase I Pain (DRG target) [39]
Retroviruses	ssRNA	8 kb	Years Integrative	Low	Infects only dividing cells	Phase III Glioma [40]
Lentiviruses	ssRNA	8 kb	Years Integrative	Low	Infects both non- and dividing cells	Phase I/II X-linked adreno-leukodystrophy [41]
Poxviruses (e.g., vaccinia)	dsDNA	>30 kbp	Weeks Replicates cytoplasm	High	Infects both non- and dividing cells	Phase III Hepatocellular carcinoma [42]

Main features of the most common viral vectors. DRG: dorsal root ganglion. Clinical trials according to the database ClinicalTrials.gov [43].

**Table 2 ijms-22-01815-t002:** Tropism of naturally occurring AAVs.

Serotype	Tropism	Reference
AAV1	CNS, Skeletal muscle	[48,49]
AAV2	CNS, Kidney, Retina	[48,50,51]
AAV3	Liver	[52]
AAV4	CNS, Heart, Lung	[53,54]
AAV5	CNS, Retina	[48,51]
AAV6	Skeletal muscle, Heart, Pancreas	[53,55,56]
AAV7	Skeletal muscle, Heart, Liver, Retina	[49,51,53]
AAV8	Skeletal muscle, Heart, Liver, Pancreas, Retina	[49,51,55,57]
AAV9	CNS, Skeletal muscle, Heart, Lung, Liver, Kidney, Retina	[51,53,58,59,60,61]
AAV10	CNS, Ileum, Lymphatic tissue	[62,63]
AAV11	Ileum, Lymphatic tissue	[62]
AAVrh10	CNS, Skeletal muscle, Heart	[64,65]

**Table 3 ijms-22-01815-t003:** Main routes of administration for human CNS gene delivery.

Delivery Route	Clinically Implemented	Procedural Risk	Efficiency Reaching		
			Spinal Cord	Cerebellum	Cerebrum
Systemic	Yes	Very low	Low	Low	Low
IT	Yes	Low	High	Medium	Low
ICM	No	High	High	High	Medium
ICV	Yes	Neonatal: Low Adult: High	Medium	Medium	High

Main advantages and limitations for each procedure.

**Table 4 ijms-22-01815-t004:** Friedreich’s Ataxia main mouse models and their major traits.

Animal Models	Specie	Tissue Frataxin or Homologue K.D./K.O.	Life Span	Genotype or Methodology	Frataxin	Example of Characteristic Phenotype Traits (Not all Could be Included for Practical Reasons)
*Fxn* K.D. (shRNA-37)	M.M.	Cerebellum	Normal	shRNA	30% of WT in the cerebellum	Motor and coordination deficiencies. [18]
MCK	M.M.	Heart	65–105 d	*Fxn* L3/E4del l3 FA allele	None in the heart, reduced in the rest of tissues.	Cardiomyopathy (Cardiac hypertrophy, failure, defective aconitase, and mitochondrial iron accumulation). [85]
NSE	M.M.	Neurons	15–33 d	*Fxn* L3/E4del l3 FA allele	None in neurons, reduced in the rest of tissues.	Neurological phenotype with age, including ataxia. Cardiomyopathy earlier than MCK. Iron accumulation. [85]
*Pvalb* cKO	M.M	Parvalbumin +	Normal (≈ 2 y)	*Fxn* L3/E4del l3 FA allele	None in Pvalb+ cells. Reduced in the rest.	Movement and coordination impairment. Sensory ataxic phenotype and neuropathy. [86]
KIKI	M.M.	Ubiquitous	2/13 < 12 m 11/13 ≈ 2 y	*Fxn* (GAA)230^+/+^	66–83% of WT levels	No pathological phenotype detected. [87]
KIKO	M.M.	Ubiquitous	Normal (≈ 2 y)	*Fxn*(GAA) 230^+/−^	25–36% of WT levels	Mild heart fibrosis [87] Deficits in mitochondrial biogenesis and respiratory chain complexes [88] and synaptic alterations [89] in the cerebellum. Neurobehavioral deficits [90].
YG22R	M.M.	Ubiquitous	Normal (≈ 2 y)	YAC *hFXN* (GAA)190	None endogenous, ≈ 60% hFXN of WT levels (cerebellum)	Mild motor and coordination defects. Slight oxidative stress. GAA instability. [9]
YG8R	M.M.	Ubiquitous	Normal (≈ 2 y)	YAC *hFXN* (GAA)90 (GAA)190	None endogenous, ≈ 60% hFXN of WT levels (cerebellum)	Mild motor and coordination defects. Slight oxidative stress. Cardiac aconitase deficient. GAA instability [9] and mitochondrial alterations [91].
YG8sR	M.M.	Ubiquitous	Normal (≈ 2 y)	YAC *hFXN* (GAA)200	None endogenous, ≈ 22% hFXN of Y47R levels (CNS)	Coordination, behavior, and motor deficits. Decreased aconitase activity. GAA instability. IFG. [10]
YG8JR	M.M.	Ubiquitous	Data not published	YAC *hFXN* (GAA)800	None endogenous, hFXN reduced	Coordination deficit (six months). Alopecia. Aconitase activity decreased (data not published).

M.M.: *Mus musculus*. IFG: impaired fasting glucose. WT: wild-type. d: days. m: months. y: years. L3: FXN conditional allele for Cre system.

**Table 5 ijms-22-01815-t005:** The history of in vivo Friedreich’s Ataxia gene therapy.

Treatment	Dose	Animal Model	Age	Delivery Route	Major Improvements	Ref.
HSV1-hFXN	1.44 × 10^4^ IU	FXN L3/L3 + HSV1-Cre	4 w after K.O.	Stereotaxic injection brain stem (intraparenchymal)	Reversal of motor coordination deficit.	[17]
AAVrh10-CAG-hFXN-HA	5.4 × 10^13^ v.g./kg	MCK	3 w	Intravenous	Prevents cardiac pathology at a cellular and molecular level.	[93]
AAVrh10-CAG-hFXN-HA	5.4 × 10^13^ v.g./kg	MCK	7 w	Intravenous	Reverses cardiomyopathy.	[93]
HSV-BDNF	4200 IU	Mouse shRNA-37	8 w	Cerebellar Cortex	Prevention of cerebellar neuropathy and ataxic phenotype.	[18]
AAV9-CAG-hFXN-HA	5 × 10^13^ v.g./kg	*Pvalb* cKO	3.5 w	Intravenous	Prevents progressive sensory defects.	[86]
AAV9-CAG-hFXN-HA + AAVrh10-CAG-hFXN-HA	5 × 10^13^ v.g./kg (AAV9) 1 × 10^10^ v.g./deposit (AAVrh10)	*Pvalb* cKO	7.5 w	Intravenous (AAV9) Striatum and Cerebellum (AAVrh10)	Rescues sensory neuropathy.	[86]
AAV9-hFXN	6 × 10^9^ and 6 × 10^11^ v.g.	MCK	5–9 d	Intraperitoneal	Cardiac systolic function better preserved over time and prolonged life.	[20]
AAV9-hFXN	Several (6 × 10^11^ v.g. highest)	NSE	5–9 d	Intraperitoneal	Prolonged life, reduced heart hypertrophy, and reversal of behavioral deficit.	[20]
AAVrh.10hFXN	10^11^ v.g.	αMyhc	6 w	Intravenous	Cardiac performance restored to control levels.	[94]

v.g: viral genomes. d: days. w: weeks. Ref: reference.

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
