# Peer review of "Future Prospects of Gene Therapy for Friedreich’s Ataxia"

_ijms, 2021, doi:10.3390/ijms22041815_

Round 1
Reviewer 1 Report
In this manuscipt Ocana-Santero and colaborators review the current state of viral vectors in gene therapy, the different Friedreich Ataxia (FA) mouse models available, and the gene therapy approaches in these FA models.
The manuscript presents a comprehensive review on the subject. Nevertheless, there are some minor points that should be addressed:
- - In table 1, reference or ClinicalTrials Identifiers should be provided for each clinical trial. Please also check the status of COVID19 trial, as an adenovirus based vaccine has already been approved
- - Please carefully proof-read to eliminate some grammatical errors.
Author Response
We thank the editor and reviewer for their constructive comments.
We have responded to each comment and amended the manuscript accordingly. Please see the attachemnt to find the point-by-point response.

Reviewer 2 Report
Ocana-Santero and co-workers provide an extensive and detailed review on gene therapy for Friedreich's Ataxia. Overall, this is a well-organized review covering the use of viral vectors in gene therapy, routes of administration for CNS delivery, mouse models of FA and the state of the art of gene therapy for FA. The authors have done rigorous reviews on the literature in the relevant fields. The manuscript cites appropriate and helpful references including review papers and cutting-edge studies so that it benefits broader readership with varying interests in gene therapy and neurologic disorders. The following minor comments are provided for the authors to further improve this manuscript.
Some of the statements in a quite a few places are not accurate. Below are just a few examples. Please carefully edit the manuscript before re-submission.
Line 102, ‘non-integrative vectors such as AAV…’ the statement is not exactly correct. AAV vectors are known to integrate at low frequency into host mammalian genomes, rAAV integration has been observed in numerous preclinical studies and even in clinical setting in patients.
In addition, the authors state that ‘AAVs are rather poor vectors for the treatment of diseases affecting highly dividing-cells such as sickle cell’, the reviewer suggest to modify this statement. Although AAV is not ideal for gene addition strategy to highly dividing cells, AAV-based CRISPR applications actually represents as a promising approach in this case, gene editing using the AAV-based CRISPR system targeting the HBB gene in CD34+ HSPCs to correct the mutation causing sickle cell disease has been reported by several groups.
The authors state that the package size of AAV is up to 4kb, this is not right. The packaging capacity of AAV is up to 4.8 kb, the two ITRs is about 0.3Kb, so the AAV system provides up to 4.5 kb to package the transgene cassette and other elements.
Author Response
We thank the editor and reviewer for their constructive comments.
Please see the attachment to find the point-by-point response.
